# Relationship between Lung and Brain Injury in COVID-19 Patients: A Hyperpolarized ^129^Xe-MRI-based 8-Month Follow-Up

**DOI:** 10.3390/biomedicines10040781

**Published:** 2022-03-27

**Authors:** Shizhen Chen, Yina Lan, Haidong Li, Liming Xia, Chaohui Ye, Xin Lou, Xin Zhou

**Affiliations:** 1State Key Laboratory of Magnetic Resonance and Atomic and Molecular Physics, National Center for Magnetic Resonance in Wuhan, Wuhan Institute of Physics and Mathematics, Innovation Academy for Precision Measurement Science and Technology, Chinese Academy of Sciences, Wuhan 430071, China; chenshizhen@wipm.ac.cn (S.C.); haidong.li@wipm.ac.cn (H.L.); ye@wipm.ac.cn (C.Y.); 2University of Chinese Academy of Sciences, Beijing 100049, China; 3Department of Radiology, Chinese PLA General Hospital, Beijing 100853, China; lanyn1220@163.com; 4Department of Radiology, Tongji Hospital, Tongji Medical College, Huazhong University of Science and Technology, Wuhan 430030, China; lmxia@tjh.tjmu.edu.cn

**Keywords:** ^129^Xe gas MRI, gas–blood exchange lung function, COVID-19 pneumonia, long-term follow-up, multiorgan involvement

## Abstract

Although the lungs are the primary organ involved, increasing evidence supports the neuroinvasive potential of severe acute respiratory syndrome coronavirus 2 (SARS-CoV-2). This study investigates the potential relationship between coronavirus disease (COVID-19)-related deterioration of brain structure and the degree of damage to lung function. Nine COVID-19 patients were recruited in critical condition from Jin Yin-tan Hospital (Wuhan, China) who had been discharged between 4 February and 27 February 2020. The demographic, clinical, treatment, and laboratory data were extracted from the electronic medical records. All patients underwent chest CT imaging, ^129^Xe gas lung MRI, and ^1^H brain MRI. Four of the patients were followed up for 8 months. After nearly 12 months of recovery, we found no significant difference in lung ventilation defect percentage (VDP) between the COVID-19 group and the healthy group (3.8 ± 2.1% versus 3.7 ± 2.2%) using ^129^Xe MRI, and several lung-function-related parameters—such as gas–blood exchange time (T)—showed improvement (42.2 ms versus 32.5 ms). Combined with ^1^H brain MRI, we found that the change in gray matter volume (GMV) was strongly related to the degree of pulmonary function recovery—the greater the increase in GMV, the higher degree of pulmonary function damage.

## 1. Introduction

At present, there is an increased awareness of multiorgan involvement in patients with coronavirus disease (COVID-19) [1,2,3]. Recent studies have shown that a considerable proportion (22–56% across different severity scales) of patients had a pulmonary diffusion abnormality 6 months after the onset of symptoms [4]. Research on persistent symptoms in the brain has shown that neurological symptoms are observed in 36.4% of patients, and are more common in severe cases (45.5%) [3]. These symptoms, including fatigue, muscle weakness, sleep difficulties and smell disorders, are clinical manifestations of neuronal damage caused by a decreased oxygen supply to the brain [5,6,7,8].

In discharged COVID-19 patients, long-term follow-up studies on persistent symptoms—such as lung function impairment and brain damage—are urgently required [9]. To date, a correlation analysis between the degree of damage to the two organs has not been reported. Conventional ^1^H MRI without ionizing radiation cannot detect pulmonary ventilation function due to the extremely low proton density in the lung parenchyma [10,11]. Hyperpolarized ^129^Xe gas MRI enables the visualization of ventilation function by boosting ^129^Xe MRI signal intensity by 4–5 orders of magnitude [12]. In this study, we aimed to explore the relationship between the changes in brain microstructure and lung function impairment using ^129^Xe gas and ^1^H MRI for the lungs and brain, respectively. 

## 2. Materials and Methods

### 2.1. ^129^Xe Lung MRI

All MRI images were acquired using a 3T human scanner (uMR780 (Xe), VerImagin, Wuhan, China). HP ^129^Xe was generated through a commercially available system equipped with a spin-exchange optical pumping unit, with a freeze-out accumulation procedure in a cold finger. The HP gas containing 2% enriched xenon (86% ^129^Xe isotope), 88% ^4^He, and 10% N_2_ was thawed using hot water and then transferred into a Tedlar bag, with approximately 9 Pa purged out using a vacuum pump. The available spin polarization of ^129^Xe gas in the Tedlar bag was calculated to be approximately 25%. The subject inhaled 1 L of the gas mixture from the functional residual capacity (FRC) to calibrate the flip angle before the imaging test. The pulmonary ventilation function, gas exchange, and microstructural parameters were then measured after the subject inhaled 1 L of gas mixture (40% xenon + 60% N_2_) from the FRC. The MRI scans were performed on each subject using a 3D bSSFP sequence with field of view (FOV) = 38 cm^2^, slice number = 20, matrix = 96 × 84, bandwidth = 800 Hz/pixel, flip angle = 12–14°, slice thickness = 9 mm, and repetition time/echo time = 5.6/2.65 ms. Both the ^129^Xe and proton MR images were reconstructed to 96 × 96 matrices to calculate the ventilation defects. The gas–blood exchange functional parameters of the lung were acquired using the HP ^129^Xe chemical shift saturation recovery (CSSR) technique with additional theoretical modeling. Herein, 21 exchange timepoints ranging from 10 ms to 700 ms were used, and the spectra were acquired using a bandwidth of 50 Hz/point with 1024 sampling points.

### 2.2. ^1^H Brain MRI

T_2_-weighted images were acquired in the transverse plane using a fast spin-echo (FSE) sequence with the following parameters: TR = 9 s, TE = 100 ms, acquisition matrix = 249 × 384, FOV = 334 × 384 mm^2^, and slice thickness = 5 mm. The volumes of gray matter (GM), white matter (WM), and cerebrospinal fluid were calculated after the images were segmented using SPM12 (The Wellcome Centre for Human Neuroimaging, UCL Queen Square Institute of Neurology, London, UK) and MATLAB (MathWorks, Natick, MA, USA). The estimated global volumes of gray matter, white matter, and cerebrospinal fluid were further normalized by correcting the intracranial volume.

### 2.3. CT Scans

All CT scans were performed on a Lightspeed 16 or VCT (General Electric Healthcare, Milwaukee, WI, USA) with patients in the supine position, and the parameters for imaging tests were as follows: tube voltage = 100–120 kVp, matrix = 512 × 512, slice thickness = 1.00–1.25 mm, and FOV = 350 mm × 350 mm. The single collimation width of the reconstructed images was 0.50–1.25 mm, and the tube current was regulated through an automatic exposure modulation system.

### 2.4. Patients

We recruited nine patients with COVID-19 in critical condition from Jin Yin-tan Hospital—the first designated hospital for patients with COVID-19 in Wuhan, Hubei, China. The patients were discharged between 4 and 27 February 2020. Demographic, clinical, treatment, and laboratory data were extracted from electronic medical records. ^129^Xe gas MRI is a uniquely capable technique for assessing ventilation, microstructure, and the gas exchange process. All patients underwent chest CT imaging, ^129^Xe gas lung MRI, and ^1^H brain MRI. Four of the patients were followed up for approximately 12 months. Healthy group data were obtained from our previous results [2]. All patients in this study were severely infected. 

The clinical classifications contain mild, moderate, severe, and critically severe symptoms, in accordance with the Chinese Clinical Guidance for COVID-19 Pneumonia Diagnosis and Treatment (eighth edition) issued by the National Health Commission of China. The discharge conditions were the same as in previously published work [2]. 

### 2.5. Statistical Analysis

We used the Shapiro–Wilk test for data distribution evaluation, whereas the 2 groups were compared with *t*-tests. All statistical tests were performed using the R language (version 4.0.2; R Foundation for Statistical Computing, Vienna, Austria). 

## 3. Results and Discussion

Four of the nine COVID-19 patients completed the study. Basic information, laboratory biochemical analysis, and CT image data of these patients are shown in Table 1. The patients’ mean age was 50.67 years old, and their average length of hospitalization was 23 days. One patient had an underlying disease—diabetes mellitus. The presenting symptoms included fever (100%), myalgia (11.1%), fatigue (33.3%), cough (55.5%), chest distress (33.3%), dyspnea (11.1%), and tachypnea (33.3%), as listed in Table 1. Lymphocytopenia occurred in two patients, while 7/9 (77.7%) of the patients had elevated concentrations of high-sensitivity C-reactive protein (> 10 mg/L). Four patients (44.4%) had increased concentrations of interleukin 6 (IL-6), and two patients (22.2%) had white blood cell counts below the normal range. All patients were treated in the isolation ward of Jin Yin-tan Hospital, and were deemed cured and discharged after an average stay duration of 23 ± 9 days (14–35 days). The lung healthiness of the patients was examined by X-ray, with no lung disease and no history of smoking found. The present study was approved by the Ethics Commission of Tongji Medical College, Huazhong University of Science and Technology, and all of the patients signed the consent forms.

Both the lungs’ microstructure and their key functional parameters can be obtained using HP ^129^Xe gas MRI at the alveolar–capillary interface where gas exchange occurs [13]. The physiological parameters, including the exchange time constant (T), septal wall thickness (d), and blood hematocrit (Hct), are shown in Figure 1. Meanwhile, the ratio of xenon signals from red blood cells (RBCs) and xenon signals from interstitial tissue/plasma (TP) at an exchange time of 100 ms were also calculated (RBCs/TP) [14,15]. Additionally, HP ^129^Xe diffusion-weighted MRI (DW-MRI) was used to derive pulmonary microstructure parameters, including acinar duct radius (R), mean linear intercept (Lm), and surface-to-volume ratio (SVR) [16]. These parameters were extracted by fitting the DW-MRI data to the anisotropic diffusion model of ^129^Xe diffusion pixel by pixel, using a nonlinear least squares algorithm. The upper part of Figure 1 illustrates the segmentation and normalization of the brain ^1^H MRI. The volumes of gray matter (GMV), white matter (WMV), and cerebrospinal fluid were calculated after the images were segmented by SPM12 using MATLAB. The estimated global volumes of gray matter, white matter, and cerebrospinal fluid were further normalized by correcting the intracranial volume.

Our previous study found that, after discharge, lung ventilation function and microstructure were relatively normal, while the gas–blood exchange function was impaired compared with that of the healthy group [2]. Cao et al. reported that after a 6-month follow-up, the proportion of participants with lung diffusion impairment was 34% (114 of 334) [4]. These results are consistent with our findings that the recovery of lung function takes an exceptionally long time. Apart from pulmonary symptoms, 36.4% of COVID-19 patients presented with neurological symptoms, according to first-hand evidence from Wuhan [4]. COVID-19 usually causes irreversible airflow limitation, which can subsequently cause decreased oxygen supply to the brain. Since the brain is susceptible to changes in arterial oxygen concentration, hypoxic stress is inevitable. In addition, patients with severe COVID-19 often experience multiple organ inflammation, which may aggravate neuronal damage [17,18,19]. Therefore, the follow-up in our study included all severe cases.

Gray matter (GM) is the region of the central nervous system that processes information, and is composed of neuronal cell bodies, dendrites, and axon terminals. Conversely, the white matter (WM) is composed of neuronal axons; thus, it does not process, but rather transmits information between gray matter and peripheral organs. Brain weight is only approximately 2% of the total body weight; however, its oxygen consumption is approximately 23% of total oxygen consumption, making it extremely sensitive to hypoxia [20]. The oxygen consumption of GM is five times higher than that of WM, which demonstrates that the tolerance of GM to hypoxia is lower [21]. Therefore, GM may be more sensitive than WM to brain edema caused by hypoxia.

The changes in lung ventilation defect percentage (VDP), gas–blood exchange time (T), brain white matter volume (WMV), and brain gray matter volume (GMV) among the four patients after discharge are shown in Figure 1. For lung structure and function, the VDP of most patients returned to normal levels (3.7%) approximately 12 months after the onset of symptoms (Figure 1a). Although T decreased over time (Figure 1b), its average amount remained significantly longer than in the healthy group (47.2 ms versus 32.5 ms). Although the patients’ lung structure returned to normal after approximately 12 months of recovery, the gas–blood exchange function did not fully recover. Regarding the changes in brain volume, the WMV increased to a certain extent in all four patients (Figure 1c), whereas the GMV showed an average decrease of 10% in three patients (patients 4, 5, and 9), and a 14% increase in only one patient (patient 8) (Figure 1d). This trend was consistent with the changes in pulmonary function parameters. The blood exchange time (T) of patients 4, 5, and 9 decreased to a certain extent (8.5–27.3 ms), whereas in patient 8 it increased by 18.6 ms.

The decreased oxygen supply associated with COVID-19 may cause neuronal damage in the brain, which manifests as clinical neurologic symptoms [5,6,7,8]. It has been reported that patients with COVID-19 may present with neurological symptomatology with repercussions on imaging examinations, and these have been described in association with ischemia focus, diffuse leukoencephalopathy, and slight gyrus rectus changes [22,23]. 

The ^1^H brain MRI, lung CT images, and ^129^Xe lung MRI of patients 8 and 9 after discharge are shown in Figure 2. On April 14, patient 9 showed abnormal lesions in the gyrus rectus and subcortical white matter of the occipital lobe bilaterally (red box area), with a slightly hyperintense T_2_ FLAIR signal. While the signal of abnormal lesions disappearing (yellow box area) on December 18 (Figure 2a). CT images showed that the flocculent ground-glass density shadow (red circle area) in the right upper lobe gradually decreased(yellow circle area) and eventually disappeared (Figure 2b). During this period, the patient’s gas–blood exchange time decreased from 64.7 ms to 36.0 ms, indicating a gradual recovery of lung function (Figure 2c). Patient 8 showed a small area of abnormal signal in the left frontal lobe, which persisted after 8 months. CT images showed that the large areas of high-density shadow in both lungs did not disappear completely; hence, the lung function did not fully recover. These results were consistent with the changes in lung function parameters measured using ^129^Xe lung MRI.

Combined with the dynamic changes in lung function parameters and brain GMV shown in Figure 1, we believe that the increase in GMV is positively correlated with the degree of lung function impairment. GMV in the increased brain regions in patients with COVID-19 had a strong positive correlation with the lung gas–blood exchange time, suggesting that the increased GMV may have resulted from brain edema caused by persistent hypoxia. Our findings hold great significance for understanding the development and the pathophysiology of COVID-19. However, our study has some limitations—especially the small cohort size. Therefore, further studies with larger cohorts are needed.

## Data Availability

The datasets generated and/or analyzed during the current study are not publicly available due to the privacy of patients, but are available from the corresponding author upon reasonable request.

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
