# Peer review of "Relationship between Lung and Brain Injury in COVID-19 Patients: A Hyperpolarized ^129^Xe-MRI-based 8-Month Follow-Up"

_biomedicines, 2022, doi:10.3390/biomedicines10040781_

Round 1

Reviewer 1 Report

This manuscript describes the study which investigates the relationship between coronavirus disease (COVID-19)-related deterioration of brain structure and the degree of damage to lung function. They extracted clinical, treatment, demographic and laboratory data of nine patients from the medical records. They used the  129Xe MRI, 1H brain MRI and chest CT imaging results of four patients and studied gas–blood exchange time (T), lung ventilation defect percentage (VDP), change in gray matter volume (GMV) and change in white matter volume (WMV) in eight months. This paper might be an inspirational idea for the other researchers in the field to investigate the parameters further. The sample size of the study is highly poor and the results cannot be significantly acceptable but overall the paper can be encouraging for the future studies.

  • Line 27: In abstract section the authors didn’t have standard deviation values of the gas-blood exchange time (T) but they mention that the values are significantly different. They cannot say that without having statistics.
  • The authors mentioned the healthy group, they never give any information about the healthy group in anywhere of the paper. There is no reference or population of the healthy group. If they use any information or numerical data from any source they need to explain and show the reference to the readers.
  • The authors claimed that their data is approximately 12 months and they used this statement in their titles, too. However all the data they have and show is from April 14 to December 18 which is 8 months gap. Instead of using approximately they need to use exact time interval that is eight months.
  • The compared the values of gas–blood exchange time (T), lung ventilation defect percentage (VDP), change in gray matter volume (GMV) and change in white matter volume (WMV) in eight months in four patients. Once they mention about four patients they could describe which patients from the Table 1. They mentioned which ones in the Figure 1. That would be much better if they clarify that the first time they mention in the paper.
  • Page 3 line 97: previous results sentence needs the references.
  • Overall the references are not enough almost any statement doesn’t have a reference some paragraphs including all medical information without a signal references. Like Page2 from line 38 to line 46. This is also similar to the other didactic paragraphs.
  • The authors should not make general statement about gas–blood exchange time (T) and change in white matter volume (WMV) data due to the standard deviation.
  • In Figure 1 there is no a,b,c and d labels on the figure.
  • Page numbers are off the paper last page is 5/9.
  • There are some typing errors such as :

disorder3

verImagin ( somewhere uppercase)

China6

Figure 2 is not bold in multiple places

Reviewer 2 Report

Chen and colleagues present an interesting article on lung and brain injury in COVID-19. Before acceptance, some aspects need to be considered:

  - line 40: typo “3” -design: the study-design remains unclear: was the study conducted retrospectively or prospectively? Wording suggests prospectively (“recruited”). Please clarify. If the study was actually conducted prospectively, this is a benefit and should be named in the abstract. Did you perform a power analysis (for the prospective study)? -Why did you include 9 patients for analysis? As a result from the power analysis? - line 100: typo “6” -When you present data as mean (such as age) this display implies normal distribution. Did you check for normality? If so, please state your analysis in the methods section. If data are not normally distributed they should be presented as median + IQR. -Table 1: typo - you probably intend to use “mL” instead of “ML” -no legend is provided to table 1 -You could indicate which patients are the 4 patients used for analysis when showing data in table 1 -Please elaborate on the limitations of your study -Please elaborate on the critical discussion of your findings: could it be, that both observations are recorded independent of each other and no “true” correlation may be inferred from your findings

Author Response

Response to Reviewers

Manuscript ID: Biomedicines-1622723

We thank the reviewers for carefully reading the manuscript and for providing insightful suggestions. We have addressed all concerns noted, as detailed in the response letter below. The new material is included in the main text and recopied in the letter below for convenience. The revised manuscript is significantly improved as a result.

Chen and colleagues present an interesting article on lung and brain injury in COVID-19. Before acceptance, some aspects need to be considered:

  1. - line 40: typo “3” -design: the study-design remains unclear: was the study conducted retrospectively or prospectively? Wording suggests prospectively (“recruited”). Please clarify. If the study was actually conducted prospectively, this is a benefit and should be named in the abstract. Did you perform a power analysis (for the prospective study)? -Why did you include 9 patients for analysis?

Response: Yes, this is a prospective study, but the sample size is very small, only 9 patients. There are two reasons for the small sample size of our study. First, the 129Xe MRI for human lung, developed independently by our team, is the only hyperpolarized gas MRI equipment that has entered the clinical application in the world. So, our clinical research using the equipment has just started. Another reason is that nine patients dropped out of the study halfway due to the long follow-up time. We acknowledge a certain degree of uncertainty in our results due to the small sample size. So, we are very careful in what we conclude in our manuscript and hope the specific points we raise can motivate the further clinical application of 129Xe MRI and more research directions.

  1. As a result from the power analysis? - line 100: typo “6” -When you present data as mean (such as age) this display implies normal distribution. Did you check for normality? If so, please state your analysis in the methods section. If data are not normally distributed, they should be presented as median + IQR.

Response: As suggested by the reviewer, we have checked the normality of data presented as mean using the Shapiro-Wilk test. The results showed that age and days of hospitalization for 9 patients, and T, WMV and GMV for 4 patients all obey normal distribution. We have also added a section to Materials and Methods named ‘2.5. Statistical analysis’ in the revised manuscript (page 3, line 100).

2.5. Statistical analysis

We used Shapiro-Wilk test for data distribution evaluation, whereas the 2 groups were compared with t-test. All statistical tests were performed using R software (version 4.0.2).

  1. Table 1: typo - you probably intend to use “mL” instead of “ML” -no legend is provided to table 1 -You could indicate which patients are the 4 patients used for analysis when showing data in table 1 -Please elaborate on the limitations of your study -Please elaborate on the critical discussion of your findings: could it be, that both observations are recorded independent of each other and no “true” correlation may be inferred from your findings.

Response: Thanks for your suggestion. In the revised manuscript, we describe which patients from the Table 1 when we first mentioned these four patients. We also described the four patients in the figure 1 corresponding to the patients in Table 1 in the figure caption. We also carefully checked the whole manuscript and corrected the other misnomers.

Round 2

Reviewer 1 Report

Thank the authors for their revised manuscript. I would like to accept the article in this present form.